# The Relationship between Smartphone Addiction, Parent–Child Relationship, Loneliness and Self-Efficacy among Senior High School Students in Taiwan

**Yao-Chung Cheng** [1,*] **, Tian-Ai Yang** [2] **and Jin-Chuan Lee** [3]

1   Center for Teacher Education, National Changhua University of Education, Changhua 500, Taiwan
2   Department of Guidance and Counseling, National Changhua University of Education, Changhua 500, Taiwan; M0811018@gm.ncue.edu.tw
3   Center for Teacher Education, Southern Taiwan University of Science and Technology, Tainan 710, Taiwan; jclee@stust.edu.tw
*   Correspondence: yaochung@cc.ncue.edu.tw; Tel.: +886-7232105 (ext. 1227)

**Abstract:** This study explores the link between smartphone addiction in senior high-school students, parent–child relationship, loneliness, and self-efficacy on the basis of the theory of planned behavior (TPB) and social cognitive theory (SCT). A survey of 2172 students (1205 female students, 966 male students; mean age = 16.58 years, SD = 0.78) from 32 senior high schools in Taiwan was conducted. Moderation mediation analysis was performed using Model 14 of SPSS PROCESS-macro to test the hypotheses of this study. The result showed that the parent–child relationship was negatively related both to smartphone addiction and loneliness, which mediated the link between parent–child relationship and smartphone addiction. Self-efficacy was also found to moderate the level of loneliness related to smartphone addiction. Specifically, loneliness will ease when the parent–child relationship improves, and smartphone addiction will accordingly lessen. It was also discovered that the elevation of self-efficacy could mitigate the level of addiction. Lastly, this study provided parents, education agencies, and other policymakers in the education sector with implications based on these findings. Preventive measures for smartphone addiction and recommendations for future investigations are also given.

**Keywords:** smartphone addiction; parent–child relationship; loneliness; self-efficacy

## 1. Introduction

Smartphones have penetrated into people's lives at a speed that one can barely notice. Problematic use of smartphones has been differentiated as a type of addiction [1]. Due to the adverse impact of smartphone addiction on affect, social, and behavioral development [2,3], it has become one of the most common non-drug addictions in the contemporary era. On top of this, smartphone addiction shares typical withdrawal symptoms associated with drug, internet, and other types of addiction [4], such as being out of control, tolerance, mood changes, and relapse [5,6]. Reactions resulting from brain activation by certain stimuli are also identical [7]. In particular, smartphone addiction has a significant impact on adolescents, resulting in a number of mental and physical dysfunctions such as suicide, sleep disorder, anxiety, depression, unhappiness, and low self-esteem [6,8–11]; in addition, it spoils their academic performance [12–15] and leads to interpersonal difficulties [16,17].

It is evident that teenagers and young adults are heavier users of smartphones [4]. Smartphone addiction has translated into a serious behavioral issue among the young. The considerable amount of time that they spend on smartphones influences their interpersonal difficulties [18]. Conversely, a positive parent–child relationship can improve social skills and reduce problematic behaviors [19,20]. In particular, loneliness is a common emotional response among adolescents with poor parent–child interaction [21,22].

Antognoli-Toland [22] found that the parent–child relationship was one of the best predictors of adolescent loneliness; namely, a poorer parent–child relationship fosters a greater level of loneliness in young people. Loneliness in today's world seems increasingly ubiquitous [23], while the use of communication devices in the era of the Internet has made this loneliness worse [24]. In addition, Zhang et al. [25] saw that the parent–child relationship could indirectly affect teen Internet addiction, via associated loneliness. In contrast, people with high self-efficacy tend to take more positive action to offset their loneliness [26,27].

It is students in senior high schools who are most susceptible to smartphone addiction, as extensive studies have found [8,28]. In this connection, this research investigates senior high-school students in Taiwan to explore links between parent–child relationship, loneliness, self-efficacy, and smartphone addiction. The aim is to provide an empirical basis for prevention and counseling intervention regarding smartphone addiction. Suggestions and preventive measures against smartphone addiction based on the results of this study will be shared with parents, education agencies, and policymakers in the education sector.

## 2. Theoretical Background and Research Hypotheses

### 2.1. Theoretical Background

The theory of planned behavior (TPB) [29] argues that either the intentions or behaviors of people can be manipulated by attitudes, subjective norms, and perceived behavioral control, in order to target behaviors [29–31]. The TPB model has been widely applied in the area of technology, such as in the use of mobile phones [32–34] or in health-behavior related studies [35,36]. Similarly, social cognitive theory (SCT) [37] features self-efficacy and outcome expectations in relation to behavioral intentions [38,39]. Bandura believed that personal behaviors were the outcome of the undisrupted interaction between a person (or an individual) and their surrounding environment; personal behaviors are formed by personal factors interacting with external environments. This study, under the framework of TPB and SCT, elaborates on factors that influence senior high-school students' use of their mobile devices. Their behavioral intentions regarding their phones are driven by attitudes, norms, and perceived behavioral control [40], while both parent–child relationship, loneliness, and self-efficacy are important factors that influence smartphone addiction. Figure 1 depicts the theoretical model of our study, and each path of this model is explained below.

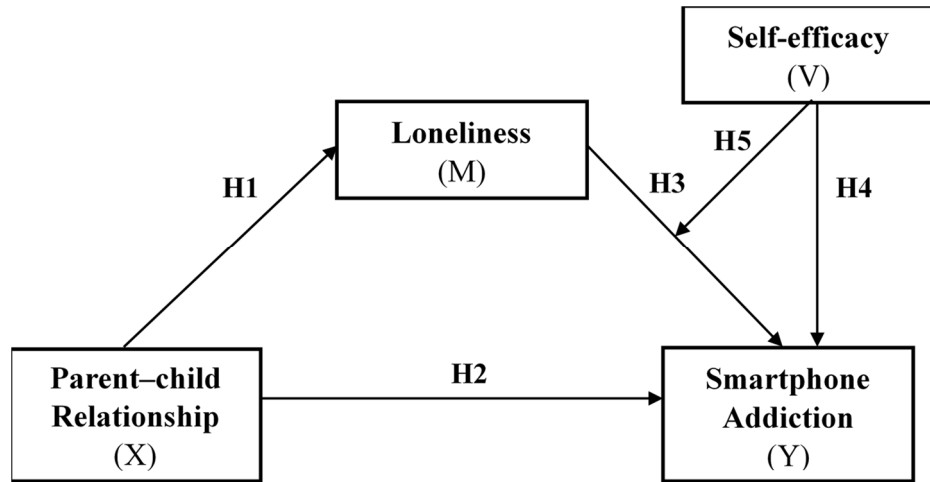

**Figure 1.** Model diagram.

### 2.2. Research Hypotheses

2.2.1. Association between Parent–Child Relationship and Loneliness

Loneliness was regarded as a pivotal index of mental health. It was a negative emotional condition and could affect a person's brain [26,41]. Loneliness was associated with poor social support and a high mortality rate [42]. In particular, loneliness found in

adolescents might lead to a higher risk of death and lower income as they came to middle age [43]. Loneliness was a result of the absence of societal integration, making young people feel either ostracized or marginalized, and pain, compounded with anxiety, would follow [44]. In addition, whether the family structure was complete or not was associated with loneliness: an adequate family environment was helpful to teenagers and young adults in dealing with loneliness [45]. If their family connections were unsatisfactory, they could have a higher risk of feeling lonely, and loneliness could spiral over time [46]. In this connection, we posit that

**Hypothesis 1 (H1).** *Parent–child relationship is negatively associated with loneliness.*

### 2.2.2. Association between Parent–Child Relationship and Smartphone Addiction

Liu and Kuo [47] found that the parent–child relationship could influence both the social support that parents provided to their children and the establishment of a sense of security; a poor parent–child relationship could lead children to seek other channels for the desired support they could have received from their parents, such as via the Internet [48]. In particular, family dysfunction could increase the risk of teen smartphone addiction [49,50] as the parent–child relationship acted as a vital catalyst for smartphone addiction and was negatively related to it [51–53]. Other studies also showed that when the communication between parents and children was carried out through smartphones, it could worsen the parent–child relationship and exacerbate the smartphone addiction of adolescents [54]. Various parenting styles could prompt children to act differently in terms of the use of smartphones [55], while the parent–child relationship could negatively predict teen smartphone addiction [56,57]. Given the aforementioned literature and TPB, this study argues that the subjective perception of senior high-school students of their parent–child relationship might affect their use of smartphones. We propose the following hypothesis.

**Hypothesis 2 (H2).** *Parent–child relationship is negatively associated with smartphone addiction.*

### 2.2.3. Association between Loneliness and Smartphone Addiction

Some studies discovered that loneliness among college students could serve as a predictor of their compulsive Internet use and their tolerance and withdrawal reactions regarding Internet addiction. The higher level of Internet addiction became, the lonelier a person would feel [58,59]. In addition, the convenience and the mobility of mobile devices marked faster and more frequent access to social media. When an interpersonal connection was established through mobile devices, it would fuel the problematic use of mobile devices [60]. Furthermore, people who disguised themselves as someone else via the Internet could experience negative effects in the web world, making them lonelier than ever [61]. Loneliness was an internal factor in predicting perceived behavioral control that affects target behaviors. Based on the aforementioned literature and TPB, the objective judgment of senior high-school students on the internal factor, loneliness, might affect their use of smartphones. This leads to the following hypothesis.

**Hypothesis 3 (H3).** *Loneliness is positively associated with smartphone addiction.*

### 2.2.4. Association between Self-Efficacy and Smartphone Addiction

The absence of self-efficacy could undermine behavioral motivation [62], such as putting fewer efforts in academic performance and relationship maintenance, and more on smartphones. It was evident that smartphone addiction was tied to interpersonal relationship and academic difficulties. Socialization and academic self-efficacy were important factors in smartphone addiction [63]. Studies showed that self-efficacy had an association with the problematic use of mobile devices [64] as it could affect the smartphone addiction [65,66]. In particular, academic self-efficacy was negatively related to smartphone addiction, and it could mitigate the effect of smartphones on academic procrastination [67]. Given the aforementioned literature and SCT, this study argues that the

self-efficacy of senior high-school students could influence smartphone addiction. We propose the following hypothesis.

**Hypothesis 4 (H4).** *Self-efficacy is negatively associated with smartphone addiction.*

2.2.5. Self-Efficacy Mediates and Moderates Parent–Child Relationship, Loneliness and Smartphone Addiction

The quality of parent–child relationship was associated with the development of self-efficacy, and emotional support could prompt the elevation of self-efficacy [68]. In particular, the better the parent–child relationship, the more satisfied children could feel with their abilities, and the more confident they grew in achieving expected outcomes [69]. Children could feel less stressed and anxious while facing new environments, and so their self-efficacy could be improved [70]. In addition, Zhang et al. [25] found that loneliness could mediate the link between parent–child relationship and Internet addiction, while self-efficacy was related to loneliness [71]. People with higher self-efficacy were more likely to cope with loneliness better, and the chance of overcoming their loneliness was higher [72]. On top of that, a poor parent–child relationship could lead to unsatisfied psychological needs of the youth and cause them to resort to smartphones for comfort, developing smartphone addiction as a result [73]. It was clear that self-efficacy and loneliness were predictors of problematic behaviors on the Internet [74,75]. Different approaches to the use of mobile devices could depend on some important factors such as self-efficacy, parent–child relationship, and loneliness, all of which interacted with one another. Given the aforementioned literature, TPB, SCT, and all the proposed hypotheses, we posit

**Hypothesis 5 (H5).** *Self-efficacy will moderate the link between loneliness and smartphone addiction, while the indirect effect of parent–child relationship on smartphone addiction through loneliness is conditional on self-efficacy.*

## 3. Method

### 3.1. Participants and Procedure

The study population came from students who were still attending senior high schools in 2020 in Taiwan. We used convenience sampling to invite 2171 students (1205 female students, 966 male students; mean age = 16.58 years, SD = 0.78) from 32 senior high schools in Taiwan to participate in this study. Table 1 summarized the demographic information of the participants with more than half (55.5%) being female; 43.48% were from senior high schools, and 42.56% from vocational high schools. The sample consisted mainly (60.43%) of the tenth-grade students.

**Table 1.** Demographics information of the sample.

| Demographic Categories | N | Percentage % |
|---|---|---|
| Gender | | |
| *Female* | 1205 | 55.50 |
| *Male* | 966 | 44.50 |
| School Type | | |
| *Senior High School* | 944 | 43.48 |
| *Vocational High School* | 924 | 42.56 |
| *Comprehensive High School* | 303 | 13.96 |
| Grade | | |
| *Tenth grade* | 1312 | 60.43 |
| *Eleventh grade* | 465 | 21.42 |
| *Twelfth grade* | 394 | 18.15 |

Before data collection, the researcher explained the content and purpose of the research with the principals and directors of each school, principals and directors of each school were

fully informed of the aim of the research and the testing procedure to ensure the data was viable before they received the questionnaire. After their discussion, they agreed to conduct a survey in their school. Informed consent was obtained from the participants and their teachers. Teachers of each school conducted the survey and informed the participants of the confidentiality of the information collected and the principle of voluntary participation. Prior to proceeding, all the participants were not only provided with verbal and written information about how to fill in this anonymous questionnaire, but promised irrelevance of this survey to their performance. In so doing, participants can feel secured while answering questionnaires. It can also increase the accuracy rate of the data.

*3.2. Measures*

3.2.1. Description for Instruments

- Smartphone Addiction

The Smartphone Addiction Scale administered by Jao [76] was used in this study to measure smartphone addiction of the participants. This section contained 26 items, including: (1) five items on "compulsive" with items such as "I fail to control the impulse to use smartphone"; (2) five items on "withdrawal reaction" with items such as "I feel distressed or down once I cease using smartphone for a certain"; (3) four items on "tolerance" with items such as "I find that I have been hooking on smartphone longer and longer"; (4) seven items on "interpersonal and health problems" with items such as "My interaction with family members is decreased on account of smartphone use"; (5) five items on "time management issues" with items such as "I have slept less than four hours due to using smartphone more than once". The participants were asked to rate each item on a five-point Likert scale ranging from (1) strongly agree to (5) strongly disagree. A higher score on this scale represents a higher level of smartphone addiction. In the original scale, the total variance explained is 68.153% and the Cronbach's alpha is 0.93.

- Parent–child relationship Scale

Parent–child relationship Scale administered by Chen [77] was processed in the same matter to measure the parent–child relationship of the participants. This section encompassed 18 items, which were divided into two sub-sections. They were positive aspects of the relationship, including: (1) three items on "a sense of attachment" with items such as "I hope he/she can always be by my side"; (2) three items on "a sense of admiration" with items such as "He/She has many advantages"; (3) three items on "a sense of closeness" with items such as "He/She can keenly perceive my thoughts and feelings; (4) three items on "feeling valued" with items such as "I am very important to him/her", and negative aspects, including: (5) three items on "negative affect" with items such as "It is easy to get angry when I am with him"; (6) three items on "the lack of self-autonomy" with items such as "I often feel that he/she disciplines me too much". The items concerning the positive aspect were assessed on a five-point Likert scale (strongly agree = 1, strongly disagree = 5), while both "negative affect" and "the lack of self-autonomy" were designed with negative statements and scored reversely (strongly disagree = 5, strongly agree = 5). A higher sum in this scale indicates a closer link between parents and children. In the original scale, the total variance explained is 63.59% and the Cronbach's alpha is 0.91.

- Loneliness

The Chinese version of Russell's UCLA Loneliness scale version III [78] translated and modified for the better comprehension of respondents by Chang and Yang [79] was employed in this study. Eleven negatively worded items were for negative loneliness with items such as "How often do you feel alone?", and nine positively worded ones for positive loneliness with items such as "How often do you feel close to people?". This section adopted a four-point Likert scale instead and the measurement for positive loneliness was scored reversely (strongly disagree = 1, strongly agree = 4). According to Russell [78], loneliness measured by the UCLA Loneliness Scale is a unitary phenomenon based on

theoretical grounds. The researchers suggested that the UCLA Loneliness Scale is one-dimensional since the coefficient alphas reported for each factor were high and each factor correlated highly with the total scale score. Therefore, nine items for positive loneliness are reverse scored, whereby the higher the score, the higher level of loneliness the participants experience. In the original scale, the total variance explained is 50.6% and the Cronbach's alpha is 0.85, while the Cronbach's alpha here is 0.85.

- Self-efficacy

Self-efficacy scale for high school students administered by Kuo [80] was used to test the level of self-efficacy of senior high-school students. This section encompassed 25 items, including six items on "academic performance" with items such as "I scheduled my study time"; six items on "career development" with items such as "I know the direction of my future"; seven items on "interpersonal relations" with items such as "I am good at communicating with friends"; six items on "physical performance" with items such as "I exercise regularly". The participants rated each item on a five-point Likert scale ranging from (1) strongly agree to (5) strongly disagree. A higher score shows a higher level of self-efficacy with the participants. In the original scale, the total variance explained is 61.30% and the Cronbach's alpha is 0.90.

### 3.2.2. Confirmatory Factor Analysis for Instruments

The reliability and validity of the instruments were pre-tested with a randomly chosen population of 1327 senior high-school students in Taiwan for confirmatory factor analysis (see Table 2). The factor loadings on each scale were between 0.469 and 0.854, and the overall Cronbach alpha (CA) ranged from 0.91 to 0.94. Moreover, composite reliability (CR) fell between 0.498 and 0.699, variance extracted (AVE) between 0.437 and 0.645, RMSEA between 0.065 and 0.077, GFI from 0.862 to 0.923, AGFI from 0.837 to 0.902, NFI between 0.866 and 0.921, NNFI between 0.863 and 0.928, CFI between 0.875 and 0.928, as well as IFI between 0.875 and 0.928. The results indicated that the CFA of all scales was within the acceptable range [81–87].

**Table 2.** Results of confirmatory factor analysis.

| Scale Name | Factor Name | Factor Loadings | CA | CR | AVE | Fit Indices |
|---|---|---|---|---|---|---|
| **Smartphone Addiction Scale** | Compulsive | 0.515–0.796 | 0.786 | 0.551 | 0.456 | RMSEA = 0.066 GFI = 0.911 AGFI = 0.890 NFI = 0.908 NNFI = 0.906 CFI = 0.916 IFI = 0.916 Overall CA = 0.94 |
| | Withdrawal reaction | 0.676–0.780 | 0.854 | 0.624 | 0.553 | |
| | Tolerance | 0.662–0.844 | 0.847 | 0.616 | 0.542 | |
| | Interpersonal and health problems | 0.581–0.729 | 0.849 | 0.539 | 0.437 | |
| | Time management issues | 0.585–0.833 | 0.849 | 0.573 | 0.484 | |
| **Parent–child relationship Scale** | Positive relationship | 0.654–0.805 | 0.914 | 0.662 | 0.536 | RMSEA = 0.077 GFI = 0.912 AGFI = 0.884 NFI = 0.913 NNFI = 0.904 CFI = 0.918 IFI = 0.918 Overall CA = 0.91 |
| | Negative relationship | 0.608–0.836 | 0.867 | 0.583 | 0.499 | |
| **UCLA Loneliness Scale Version 3** | Positive loneliness | 0.469–0.694 | 0.851 | 0.498 | 0.381 | RMSEA = 0.065 GFI = 0.923 AGFI = 0.902 NFI = 0.915 NNFI = 0.912 CFI = 0.923 IFI = 0.923 Overall CA = 0.94 |
| | Negative loneliness | 0.518–0.802 | 0.891 | 0.555 | 0.461 | |
| **Self-efficacy scale for high school students** | Academic performance | 0.690–0.829 | 0.844 | 0.605 | 0.528 | RMSEA = 0.066 GFI = 0.904 AGFI = 0.884 NFI = 0.921 NNFI = 0.920 CFI = 0.928 IFI = 0.928 Overall CA = 0.91 |
| | Career development | 0.652–0.860 | 0.914 | 0.692 | 0.645 | |
| | Interpersonal relations | 0.676–0.854 | 0.902 | 0.640 | 0.574 | |
| | Physical performance | 0.784–0.843 | 0.918 | 0.699 | 0.652 | |

Note: CA = Cronbach's alpha; CR = composite reliability; AVE = average variance extracted.

### 3.3. Analytical Strategy

The Model 14 of PROCESS-macro, or the second stage of the moderation model, (version 3.5) by Hayes [88] was used in this study [89]. This model utilized ordinary least squares regression to generate model coefficients and estimate the direct and indirect effect of the mediating mechanism [90]. Figure 2 illustrated the effect of the predictor variable (parent–child relationship, X) on the mediator variable (Loneliness, M), impacting the outcome variable (smartphone addiction, Y). On the other hand, the pathway linking mediator variable M to the outcome variable Y was moderated by the variable V (self-efficacy). The estimated parameters of the pathways in the second stage moderation model were determined by percentile bootstrapping method, which in this study drew 5000 bootstrap samples to estimate the indirect effect. This method was suggested by Hayes and Scharkow [91] to examine the indirect effect in this scenario for being as the best compromise among various statistics methods in terms of power and type I error. The indirect effect will be considered significant when the 95% confidence intervals do not include or cross zero [90].

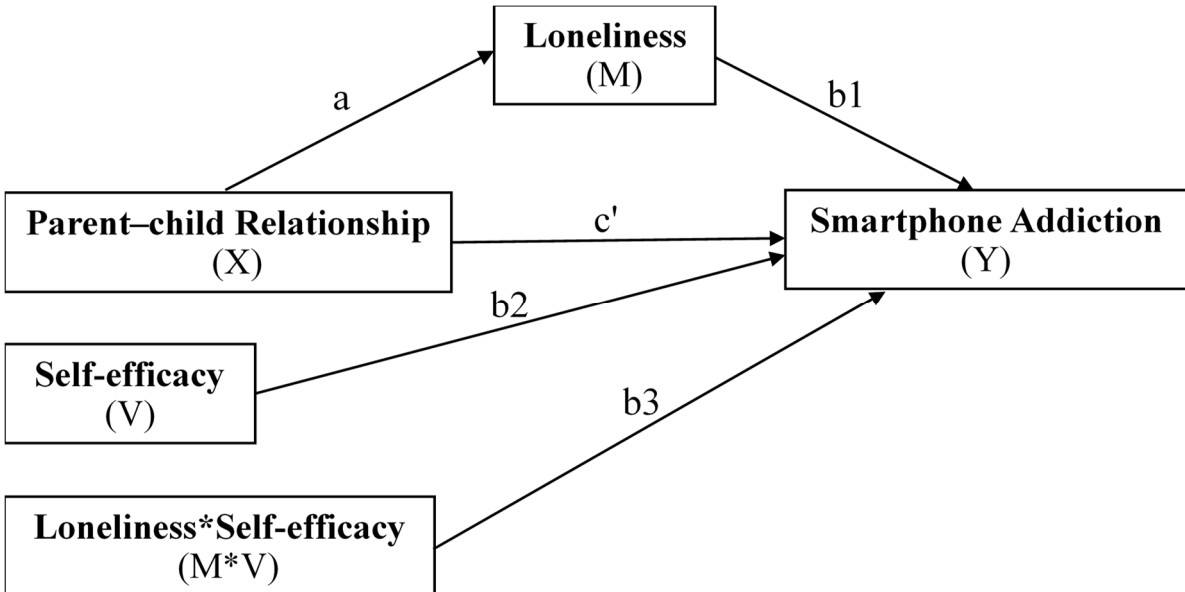

**Figure 2.** Second stage moderation model.

## 4. Results

### 4.1. Test the Extent of the Common Method Variance (CMV)

The collected data came from distributing hard copies of the questionnaire to different senior high schools in the same period of time, so CMV might be a problem [92]. For the validity of the analysis, we examined the potential threat of CMV with two different methods. First, Table 3 showcased that co-relation values of all the constructs in the study were less than 0.90, meaning the absence of CMV [93]. Second, Harman's single-factor test was used to test all the 89 items of the four scales for the extent of the CMV via principal-component factor analysis [94]. The result saw that the eigenvalue of 16 factors was more than 1, or 64.07%, with the first one accounting only for 17.99%, far away from the cutoff value dividing large or small CMV of 40% [95]. It was clear that the co-variance between independent variables and dependent variables depended largely on the constructs rather than the measurement methods used during the data collection.

**Table 3.** Validity and descriptive statistics of each variable.

| Variable | Mean | *SD* | X | M | V | Y |
|---|---|---|---|---|---|---|
| Parent–child relationship (X) | 3.510 | 0.657 | 1 | | | |
| Loneliness (M) | 2.371 | 0.570 | −0.305 *** | 1 | | |
| Self-efficacy (V) | 3.408 | 0.605 | 0.303 *** | −0.480 *** | 1 | |
| Smartphone Addiction (Y) | 2.487 | 0.678 | −0.179 *** | 0.245 *** | −0.181 *** | 1 |

*** $p < 0.001$.

### 4.2. Correlation Analysis and Descriptive Statistics

Pearson's correlation analysis was conducted to ensure the correlation between all the variables in the moderated mediation mechanism [90]. The descriptive statistics in Table 3 showcased that all the variables (including predictor variable X, mediator variable M, moderator variable V, and outcome variable Y) were significantly correlated ($p < 0.001$). Predictor X was negatively correlated with mediator M and variable Y, mediator M and moderator V, and moderator V and variable Y. On the other hand, predictor X and moderator V was positively correlated with mediator M and variable Y.

### 4.3. Estimating Parameters of the Moderated Mediation model

In this study, parent–child relationship (X) was an independent variable, smartphone addiction (Y) was an outcome variable, loneliness (M) was a moderator variable, self-efficacy (V) was a mediator variable in the second stage mediation model. Table 4 showcased that parent–child relationship (X) could significantly and positively predict loneliness (M) (a = −0.264, t = −14.901, $p < 0.001$) as that explained 9.3% of the total variance, supporting Hypothesis 1. Furthermore, either parent–child relationship (X), loneliness (M), self-efficacy (V), or loneliness x self-efficacy (M*V) could significantly predict smartphone addiction (Y) (c′ = −0.103, t = −4.533, $p < 0.001$; b1 = 0.238, t = 8.241, $p < 0.001$; b2 = −0.065, t = −2.410, $p < 0.05$; b3 = 0.153, t = 4.008, $p < 0.001$) as that explained 8.2% of the total variance. On the other hand, the total variance explained for the interaction of loneliness x self-efficacy (M*W) is 0.7%. This supports Hypothesis 2, Hypothesis 3 and Hypothesis 4.

**Table 4.** Parameter estimation of the moderated mediation effect.

| | | **Model 1** | | | | | **Model 2** | | | |
|---|---|---|---|---|---|---|---|---|---|---|
| **Outcome Variable** | | **Loneliness (M)** | | | | | **Smatphone Addiction (Y)** | | | |
| **Predictor** | | *B* | *SE* | *t* | | | *B* | *SE* | *t* | |
| Constant | | 0.928 | 0.063 | 14.647 | *** | | 2.875 | 0.081 | 35.484 | *** |
| Parent-child Re. (X) | a | −0.264 | 0.018 | −14.901 | *** | c′ | −0.103 | 0.023 | −4.533 | *** |
| Lonelinss(M) | | | | | | b1 | 0.238 | 0.029 | 8.241 | *** |
| Self-efficacy (V) | | | | | | b2 | −0.065 | 0.027 | −2.410 | * |
| M*V | | | | | | b3 | 0.153 | 0.038 | 4.008 | *** |
| Model summary | $R^2 =$ | 0.093 | $F_{(1.2169)} =$ | 222.035 | *** | $R^2 =$ | 0.082 | $F_{(4.2166)} =$ | 48.119 | *** |
| | | | | | | Interaction $\Delta R^2 =$ | 0.007 | $F_{(1.2166)} =$ | 16.062 | *** |

* $p < 0.05$, *** $p < 0.001$.

### 4.4. Testing Each Effect of the Moderated Mediation Mechanism

Direct effect and conditional indirect effect were analyzed with the Model 14 of PROCESS-macro 3.5 [88] as Table 5 showed. First, the direct effect of parent–child relationship (X) on smartphone addiction (Y) was significantly negative (c′ = −0.103, t = −4.533,

$p < 0.001$); namely, parent–child relationship (X) was significantly and negatively related with smartphone addiction (Y). This suggested that this model was partial mediation effect. Second, the conditional indirect effect of parent–child relationship (X) on smartphone addiction (Y) through loneliness (M) with self-efficacy as the second-stage moderator in the mediation model was significantly negative. Specifically, (1) when self-efficacy (V) was low, the conditional indirect effect of parent–child relationship (X) on smartphone addiction (Y) through loneliness (M) was significant a * (b1 + b3 * V) = −0.038, BootCI = (−0.058, −0.020); (2) when self-efficacy (V) was medium, the conditional indirect effect was also significant a * (b1 + b3 * V) = −0.063, BootCI = (−0.083, −0.045); (3) same as self-efficacy (V) was high a * (b1 + b3 * V) = −0.087, BootCI = (−0.116, −0.060).

**Table 5.** The conditional indirect effect of moderator.

| Type (Coefficient) | Effect Path | Estimated Effect | *T* | *p* | Bootstrapping 95% CI LL | UL |
|---|---|---|---|---|---|---|
| Direct effect(*c′*) | X -> Y | −0.103 | −4.533 | <0.001 | −0.148 | −0.059 |
| Conditional indirect effect | X ->M -> Y | A * (b1 + b3 * V) | | | | |
| low V −1*SD* | −0.604 | −0.038 | | | −0.058 | −0.020 |
| Medium V | 0.000 | −0.063 | | | −0.083 | −0.045 |
| High V +1*SD* | 0.605 | −0.087 | | | −0.116 | −0.060 |

Note: parent–child relationship = X, loneliness = M, self-efficacy = V, smartphone addiction = Y, confidence interval = CI.

Figure 3 plotted the moderating effect of the constructed self-efficacy (V) on smartphone addiction (Y) through loneliness (M); namely, high self-efficacy can attenuate the effect of loneliness (M) on smartphone addiction (Y), particularly when the loneliness (M) was low and self-efficacy was high, the level of smartphone addiction (Y) could be dropped to the lowest, supporting Hypothesis 5.

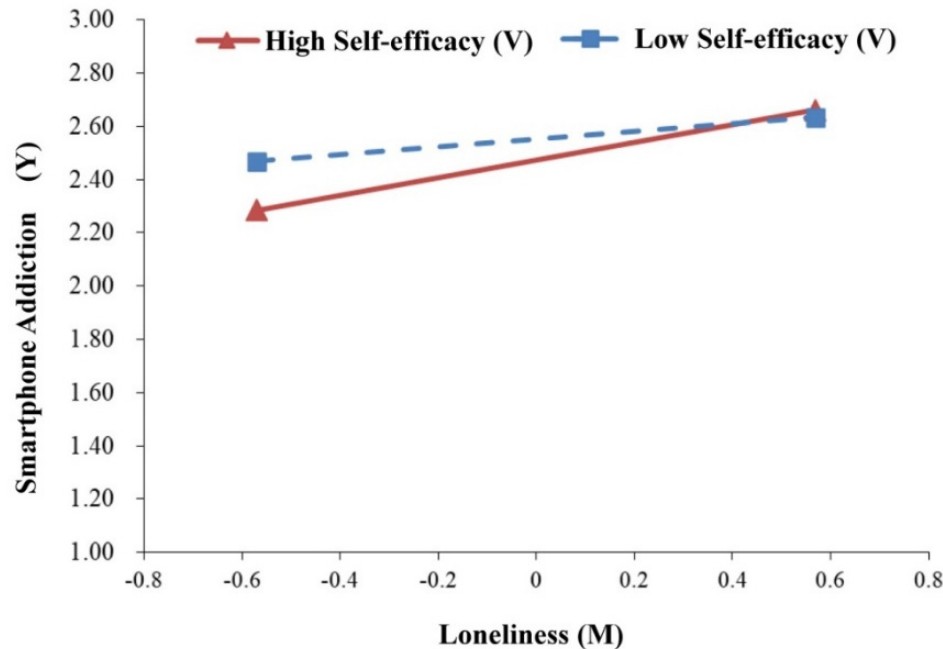

**Figure 3.** The moderating effect of self-efficacy on loneliness to smartphone addiction (y).

## 5. Discussion

### 5.1. Theoretical Implications

First, the result showed that when senior high-school students could have a better connection with their parents, they were less lonely and less addicted to their smartphones.

It fulfilled the Hypothesis 1 and Hypothesis 2 suggesting that there is an association between parent–child relationship and the decrease of loneliness and smartphone addiction. This finding is consistent with Lewin's field theory, which suggested the fundamental elements of people's behaviors were the result of undisrupted interactions between individuals and their surroundings [96]. On top of that, it was confirmed that loneliness could affect the mental health and academic performances of individuals and was related to parent–child relationship [97]. In this connection, we can confirm that loneliness can mediate the link between parent–child relationship and smartphone addiction, and the finding also coincides with the previous empirical studies. Therefore, the Hypothesis 3 was found to be supported. Furthermore, Uruk and Demir [21] found that the parent–child relationship could significantly and negatively predict teen loneliness and Internet addiction; furthermore, loneliness could predict the level of Internet addiction. In particular, other research came up with similar results and received a great number of supports from other scholars and researchers [47,98]. It is evident that loneliness can moderate the link between parent–child relationship and teen Internet addiction. Important elements constructing parent–child relationship in the study echoed some aspects in Bandura's SCT [37], such as the pattern of family interaction, and interaction quality between parents and children; on the personal factor, it included human cognition and other influences where loneliness was one of them. Parent–child relationship can mitigate loneliness and further affect addictive smartphone behaviors.

Second, this study also discovered that self-efficacy plays a mediating role in the effect of loneliness on smartphone addiction. In particular, high self-efficacy can attenuate the effect of loneliness on smartphone addiction; namely, a high self-efficacy and low loneliness can foster the lowest level of smartphones addiction which satisfied the Hypothesis 4 and Hypothesis 5 of the study. Such a finding leads us to seeing an implication that self-efficacy and loneliness could impact smartphone addiction. Similarly, Yang et al. [99] found that adolescents with high self-efficacy could feel less lonely. These two variables were negatively related. This result aligns with Cudo et al. [100]. Their findings suggested that both loneliness and self-efficacy could completely mediate the link between personal distress and problematic video gaming. This finding echoed Bandura's self-efficacy theory as Bandura argued that the intensity of behavioral motivation of individuals depended on personal assessment on self-efficacy. In short, high self-efficacy can contribute to low negative feelings of individuals. Our findings are consistent with this argument as high self-efficacy can foster low loneliness and reduce the level of addictive smartphone behaviors.

*5.2. Practical Implication*

The number of mobile-phone users in Taiwan in April 2020 reached 29.06 million, or 125% of the total population [101]; young adults aged between 18 and 19 who used mobile devices on average have topped 5.2 hours a day [102]. Smartphone addiction has become one of the most urgent issues, and particularly teenagers suffer the most [4]. In 2011, the Ministry of Education of Taiwan (MOE) published a guideline to mobile device usage on campuses as a reference for schools to respond to this smartphone epidemic. All the secondary schools can accordingly formulate a set of regulations and rules over how to use mobile devices correctly, fostering a good habit of students using electronic devices [103]. The following are some practical implications for parents, educators, and policymakers to cultivate a healthy and correct mindset of smartphone usage in children.

On one hand, in regard to family interaction, parents should make more time for their children and commit themselves to create quality family time together as the parent–child relationship is the important element of addictive smartphone behaviors found in children. In this connection, by understanding what their children really desire, and by accompanying them on their own terms, parents can effectively reduce their children's loneliness and aid children to develop multiple interests for the elevation of their self-efficacy, rather than focus solely on their academic performances.

On the other hand, teachers should encourage students to interact with their peers more for avoiding a deficiency of interpersonal connection, which could trick students into feeling lonelier. Meanwhile, teachers should have a further understanding of students' relationships with their parents so as to offer the most-fitting advice to students on how to ask for more quality family time with their parents. During the class, teachers should guide their students to navigate academic, career, and physical obstacles for a sense of achievement and fulfilling experiences by supporting them all the way to the end. Students can, therefore, grow a higher level of self-efficacy, and thus be less addicted to smartphones.

Lastly, policymakers should either intervene on certain levels where and when students should use their mobile devices or introduce remedies on this front. Policies should also be centered on aiding schools to address comprehensively any underlying causes of teen smartphone addiction as those causes might hide in students' daily lives. In particular, classes, parents' seminars, or other events or talks can be facilitated for the improvement of parent–child relationships. This is not only to reduce target behaviors, but the minimize student's backlashes against smartphone restrictions.

*5.3. Limitations and Scope of Future Research*

With rigorous efforts throughout the research, some limitations still exist. On the participants, the studied population is limited to senior high-school students. Other age groups are out of the scope of the study. In this connection, future studies can survey students in various learning stages to unveil any difference between each stage. On the methodology, the anonymous questionnaire only scratches the surface, as senior high-school students with difficulties, if any, could continue running away from their problems without consultation. It will be most helpful that researchers can err on the side of caution to choose wisely research participants, and to remind schools to follow up afterward. This study applies the quantitative research method to collect and analyze all the data, so future studies can use both quantitative and qualitative ones to add extra credibility to implications. On the variables, they are designed to analyze the personal factors of senior high-school students rather than to observe different participants responding to the same phenomena. Future studies can observe results with different cultures or gender difference in either parents or children and into this direction and analyze the collected data with the multilevel model for a better understanding of cross-level interactions. The study results will be more complete.

## 6. Conclusions

The study explores the effect of the relationship between senior high-school students and their parents on smartphone addiction, and how loneliness and self-efficacy play a role in it. The result saw that young people with a poor parent–child relationship can be more addicted to their smartphones, while loneliness is at play in between. Namely, the lonelier children feel, the more significant the parent–child relationship can affect their addictive smartphone behaviors. Conversely, young people with high self-efficacy can feel less lonely and less addicted to their smartphones. These findings can serve as a reference if in the future some interventions in teen problematic use of smartphones are to undertake. In terms of theoretical implications, it is about important factors of teen smartphone addiction; in terms of practical implications, it is about how to prevent it.

**Author Contributions:** Conceptualization, Y.-C.C. and T.-A.Y.; data curation, J.-C.L.; formal analysis, Y.-C.C. and J.-C.L.; funding acquisition, Y.-C.C.; investigation, T.-A.Y.; methodology, Y.-C.C. and T.-A.Y.; project administration, Y.-C.C.; resources, Y.-C.C.; software, J.-C.L.; supervision, Y.-C.C.; validation, Y.-C.C. and J.-C.L.; writing—original draft, Y.-C.C., T.-A.Y. and J.-C.L.; writing—review and editing, Y.-C.C., T.-A.Y. and J.-C.L. All authors have read and agreed to the published version of the manuscript.

**Funding:** This research was funded by the Ministry of Science and Technology in Taiwan, grant number MOST 109-2515-S-018-002 and MOST 110-2515-S-018-003.

**Institutional Review Board Statement:** Before data collection, the researcher explained the content and purpose of the research with the principals and directors of each school, and they agreed to conduct a survey in their school.

**Informed Consent Statement:** Teachers of each school conducted the survey and informed the participants of the confidentiality of the information collected and the principle of voluntary participation.

**Data Availability Statement:** All subjects were informed of the purpose and content of the study and voluntarily participated in the study. The data were collected and analyzed anonymously.

**Acknowledgments:** We are thankful to the MOST for funding (Project No. MOST 109-2515-S-018-002, MOST 110-2515-S-018-003).

**Conflicts of Interest:** The authors declare no conflict of interest.

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
