# Peer review of "The Relationship between Smartphone Addiction, Parent–Child Relationship, Loneliness and Self-Efficacy among Senior High School Students in Taiwan"

_sustainability, doi:10.3390/su13169475_

Round 1

Reviewer 1 Report

The Relationship between Smartphone addiction, Parent-children relationship, Loneliness, and Self-efficacy among senior high school students in Taiwan

 This paper explores the relationships between smartphone addiction of senior high-school students and parent-child along with some other individual differences such as loneliness and self-efficacy. The participants (N= 2,172) were senior high students.  The theory of planned behavior (TPB) and the social cognitive theory (SCT) are fostered as the theoretical framework for interpretation of the findings. The results showed that the parent-child relationship was negatively related both with smartphone addiction and loneliness, which mediated the link between parent-child relationship and smartphone addiction. On top of that, self-efficacy was found to moderate the level of loneliness in smartphone addiction. Moreover, it was also found that elevation of self-efficacy could mitigate addiction. The present study provides parents, education agencies or other education policymakers, useful information for preventive measures regarding smartphone addictions.

The paper presents a very good work following a quantitative approach which implements robust statistical methods. It follows a nice analytical strategy applying path analyses, CFA and moderation/ mediation analyses. Moreover, the Common Method Variance (CMV) issue is addressed which rarely examined.  

The paper is well written both the theoretical part and the empirical part, while the findings are clearly presented and discussed. This is really very good work and I recommend publication in this form.

Reviewer 2 Report

Please, pay attention

page 1: smartphone addiction hit adolescents harder

end of page 9: "that" is repeated two times

Reviewer 3 Report

Dear Authors,

The purpose of the job is good and shows that there is a job behind it. This must be appreciated.

However, there are aspects of the work that need to be explained in more detail, so some changes are suggested. It is explained in more detail below:

ABSTRACT:

  • Missing mean age and standard deviation of the participants.

METHOD:

Participants:

  • Missing mean age and standard deviation of the participants
  • It remains to explain whether the informed consent of the participants was obtained.

Instruments:

  • It remains to be stated if the UCLA scale is one-dimensional.
  • An example of an item can be put in each factor of each scale explained.

RESULTS:

  • The hypotheses are supported by the results. In the results it is convenient to read them without justification and in the discussion to support these hypotheses with previous literature.

DISCUSSION:

  • Justify each hypothesis, say if it is fulfilled or not, why and how.
  • Add in the limitations the possibilities of observing results with different cultures or gender difference. For example, are boys or girls more or less addicted to the telephone? Does this addiction to the relationship with parents affect girls more than boys? To a western or eastern culture?

REFERENCES:

  • Check the references.

Reviewer 4 Report

Abstract: please add the methodological procedures and concretize some of the preventive measures.

The theoretical background is nicely presented.

Method: which was sampling method applied? Why so much female students? Is this aligned with the country's demographic profile or even with the students attending school? Please explain it.

Discussion: it is truly pleasant to see a discussion section so nicely presented. It brings some novelty to the reader and the field studied.
